# Interference Suppression in EEG Dipole Source Localization through Reduced-Rank Beamforming

Eduardo Jiménez-Cruz and David Gutiérrez *

Monterrey's Unit, Center for Research and Advanced Studies (Cinvestav), Apodaca 66628, Mexico
* Correspondence: d.gutierrez@cinvestav.mx

**Abstract:** In this paper, we propose new neural activity indices for the solution of the inverse problem of localizing sources of cortical activity from electroencephalography (EEG) measurements. Such indices are based on reduced-rank beamformers, specifically the generalized sidelobe canceler (GSC), and with the purpose of suppressing the contribution of interfering sources and noise. Here, the GSC is modified with an adaptive blocking matrix (ABM) to optimally estimate and later suppress unwanted brain sources. With respect to the rank-reduction, this is achieved through the cross-spectral metrics (CSM) as they give a sense of the affinity of the beamformers' eigenstructure to the orthogonal subspace of noise an interference. Based on that, two different neural indices are proposed for the assessment of brain activation. Our realistic simulations show that a more consistent source localization is achieved through the proposed indices in comparison to the use of the traditional full-rank approach, specifically for brain sources embedded in high background activity that originates at the brain cortex and thalamus. We also prove the applicability of our methods on the localization of sources on the visual cortex produced by steady-state visual-evoked potentials.

**Keywords:** reduced-rank beamforming; cortical activity; neural indices

## 1. Introduction

Spatial filtering techniques (also known as *beamformers*) have been used in electroencephalography (EEG) and magnetoencephalography (MEG) for signal reconstruction and localization of sources of brain electrical activity. In most cases, the proposed methods are based on fitting dipole sources within a region-of-interest (ROI) which is chosen based on anatomical restrictions and assumptions about the relationships between brain sources [1]. For example, the linearly-constrained minimum variance beamformer (LCMV) introduced in [2], has been proved to be effective in localizing sources of brain electrical activity from surface recordings under the consideration that such sources have minimum correlation [3].

LCMV-based solutions and, to some extent, all source activity estimation methods based on beamforming underperform if the EEG forward model becomes ill-conditioned due to sources being closely positioned, possibly correlated, in presence of interference, or with poorly estimated signal and noise covariance matrices [3–5]. In order to overcome that, different reduced-rank estimators have been proposed, as they offer a gain in performance compared to the LCMV-solutions for estimating dipole source signals using EEG/MEG in ill-conditioned settings [6,7]. Most recently, in [8], a minimum-variance pseudo-unbiased reduced-rank (MV-PURE) estimation framework is proposed, which provides higher spatial resolution than LCMV-based solutions. However, the selection of the level of the rank-reduction is still an issue related to these methods.

The general consensus is that not a single source localization method is capable of providing high sensitivity and specificity at the same time, hence overlapping approaches might particularly be of help in the identification of sources whose activity may be relevant in specific scenarios [9]. In particular, we are interested in studying filters that are

specialized in reducing the influence of interfering sources, as they may be useful in applications where it is important to have such interferences removed from the reconstructed activity, e.g., in applications of directed connectivity measures such as partial directed coherence [10]. Hence, previous efforts of our group and collaborators have focused on reduced-rank beamformers that explicitly consider brain activity originating in regions different to those of main interest, but measured as correlated with signals of interest by the EEG sensor array (see [11] and references therein). The reduced-rank approach provides a significant gain in performance in such settings, as it introduces a small amount of bias in exchange for large savings in variance [12]. Our group have also previously proposed a technique in which additional restrictions on the ROIs are considered based on the *sparsity* of the neural activity [13], and such an approach is inspired in the rank-reduction provided by the *eigencanceler* [7].

Adding up to the listed efforts, in this paper we present an approach based on the structure of the generalized sidelobe canceler (GSC) [14]. The proposed source localization methods are based on the optimization of *neural indices* that are designed to reduce the contribution of the noise and interference components to the output of the GSC. This idea was initially explored in [15] but in its *quiescent* (non-adaptive) form, which is known to be LCMV-optimal only if the interference signals arrive from specific directions (see [16] for details). Instead, we propose the use of a modified structure in which signal components are adaptively selected from the noise subspace of the GSC in order to select the optimal rank-reduction required for effective interference suppression [17].

## 2. Materials and Methods

In terms of the methods, here we introduce our measurement model based on EEG data and we explain the basis under which our proposed brain activity indices are derived from the structure of the GSC. Since much of our evaluation will be conducted in terms to comparisons to other well known activity indices, we review them as well. Before we present the methods in full detail, Table 1 provides a summary of variables and notations used throughout this paper.

With respect to the materials, in this section we explain the way in which realistic EEG data is simulated. Finally, we briefly describe the database that was used in order to show the applicability of our proposed methods with real EEG data.

**Table 1.** Summary of variables and notations used in this paper (arranged in appearance order).

| Variable/Notation | Description |
|:---:|:---:|
| $\boldsymbol{q}_l$ | $l$th dipole source |
| $L$ | number of dipoles |
| $Q$ | matrix containing all dipole sources |
| $q_{l,x}(t), q_{l,y}(t), q_{l,z}(t)$ | time-varying magnitudes of $l$th dipole's Cartesian components |
| $N$ | total number of time samples |
| $\boldsymbol{r}$ | vector representing a position (Cartesian coordinates) |
| $\Omega$ | volume of the brain |
| $\boldsymbol{r}_l$ | $l$th dipole's position |
| $\boldsymbol{\theta}$ | matrix containing all $L$ dipole's positions (parameter of interest) |
| $y_m(t)$ | time-varying EEG measurement at the $m$th sensor |
| $M$ | total number of sensors |
| $Y_k$ | matrix with all EEG measurements at the $k$th experiment (trial) |
| $K$ | total number of independent trials |
| $A_l(\boldsymbol{r}_l)$ | $l$th lead field matrix associated to the $l$th dipole's position |
| $A(\boldsymbol{\theta})$ | matrix comprising the $L$ lead field matrices as a function of $\boldsymbol{\theta}$ |
| $v_m(t)$ | measurement noise realization in the $m$th sensor at time $t$ |
| $\sigma_{\mathrm{v}}^2$ | variance of measurement noise |
| $V_k$ | matrix with all measurement noise at the $k$th trial |
| $\widehat{Z}$ | indicates a consistent estimate of $Z$ |

**Table 1.** *Cont.*

| Variable/Notation | Description |
|---|---|
| $W$ | spatial filter (beamformer) |
| $I$ | identity matrix |
| $\mathbf{0}$ | matrix full of zeros |
| $\widehat{Q}$ | indicates an approximation of $Q$ |
| $\widetilde{Q}_k$ | approximated dipoles at the $k$th trial |
| $A(\boldsymbol{r})$ | lead field matrix as a function of $\boldsymbol{r}$ |
| $W(\boldsymbol{r})$ | beamformer as a function of $\boldsymbol{r}$ |
| $W_{\mathrm{LCMV}}$ | linearly constrained minimum variance (LCMV) beamformer |
| $R$ | data covariance matrix |
| $P$ | noise covariance matrix |
| $\iota_{\mathrm{LCMV}}(\boldsymbol{r})$ | LCMV-based neural activity index (NAI) as a function of $\boldsymbol{r}$ |
| $\iota_{\mathrm{MAI}}(\theta)$ | multi-source activity index (MAI) as a function of $\boldsymbol{\theta}$ |
| $G(\theta)$ | reciprocal of the noise power as a function of $\boldsymbol{\theta}$ |
| $H(\theta)$ | reciprocal of the sources' power as a function of $\boldsymbol{\theta}$ |
| $W_{GSC}$ | generalized sidelobe canceler (GSC) |
| $W_h$ | quiescent component of the GSC |
| $W_0$ | noise-plus-interference components of the GSC |
| $C_\perp$ | blocking matrix |
| $P_A^\perp$ | projection matrix of $A$ |
| $X_0$ | undesired measurement components |
| $R_{X_0}$ | autocorrelation matrix of the undesired signals |
| $\mathbf{b}$ | Wiener filter that minimizes the mean-squares of $X_0$ |
| $Q_0$ | matrix containing all undesired signals |
| $\iota_{\mathrm{RR}_1}$ | first proposed reduced-rank (RR) NAI a function of $\boldsymbol{\theta}$ |
| $\widetilde{W}_0$ | RR approximation of $W_0$ |
| $\lambda_j$ | eigenvalues of $R_{X_0}$ |
| $\boldsymbol{u}_j$ | orthonormal eigenvectors of $R_{X_0}$ |
| $\iota_{\mathrm{RR}_2}$ | second proposed RR-NAI as a function of $\boldsymbol{\theta}$ |
| $R_{X_0 Q_h}$ | cross-correlation of $X_0$ and $Q_h$ |
| $\boldsymbol{\theta}_\Delta$ | matrix with all positions of interference sources |
| $A(\boldsymbol{\theta}_\Delta)$ | lead field matrix as a function of $\boldsymbol{\theta}_\Delta$ |
| $Q_\Delta$ | matrix containing all interference dipole sources |
| $\sigma_\Delta^2$ | variance of biological noise |
| $\eta_m$ | signal-to-measurement-noise ratio |
| $\eta_b$ | signal-to-biological-noise ratio |
| $\boldsymbol{\theta}_{\mathrm{cand}}$ | matrix containing candidate dipole's positions |
| $b_{l,k}$ | bias of the estimate of $\boldsymbol{r}_l$ at the $k$th trial |
| $\mathrm{SS}_k$ | sum-of-squares of $b_{l,k}$ at the $k$th trial |
| $b_{\mathrm{MAX},k}$ | maximum bias at the $k$th trial |
| $\overline{\mathrm{SS}}$ | average sum-of-squares |
| $\sigma_{b_{\mathrm{MAX}}}$ | standard deviation of the maximum bias |
| $P(W_0)_{\mathrm{min}}$ | minimum power of $W_0$ |

## 2.1. Data Model

EEG data is assumed to be generated by cortical sources that can be approximated by $l = 1, 2, \ldots, L$ equivalent current dipoles (ECDs), whose magnitude is given by $\boldsymbol{q}_l = [q_{l,x}(t), q_{l,y}(t), q_{l,z}(t)]^T$ (assuming a Cartesian coordinate system), for $t = 1, 2, \ldots, N$ time samples, and located within the brain. The dipoles are allowed to change in time but remain at the same position $\boldsymbol{r}_l$ during the measurements period. Note that the ECD approximation is valid in practice for evoked response and event-related experiments [18].

Under those conditions, we can define a spatio-temporal representation of our EEG measurements through a matrix $Y_k$ of size $M \times N$ at the $k$th trial, for the case of $k = 1, 2, \ldots, K$ independent experiments (trials), such that

$$Y_k = \begin{bmatrix} y_1(1) & y_1(2) & \cdots & y_1(N) \\ y_2(1) & y_2(2) & \cdots & y_2(N) \\ \vdots & \vdots & & \vdots \\ y_M(1) & y_M(2) & \cdots & y_M(N) \end{bmatrix}, \tag{1}$$

where $y_m(t)$ is the time-varying EEG signal at the $m$th sensor, for $m = 1, 2, \ldots, M$. Based on that, the following measurement model can be proposed:

$$Y_k = A(\boldsymbol{\theta})Q + V_k, \tag{2}$$

where $A(\boldsymbol{\theta}) = [A_1(\boldsymbol{r}_1) \cdots A_L(\boldsymbol{r}_L)]$ is the $M \times 3L$ *lead field* matrix containing the individual $A_l(\boldsymbol{r}_l)$ responses to each dipole source, $\boldsymbol{\theta} = [\boldsymbol{r}_1, \ldots, \boldsymbol{r}_L]^T$ represents the positions of the sources (parameter to be estimated), $Q = [\boldsymbol{q}_1, \ldots, \boldsymbol{q}_L]^T$ is the $3L \times N$ matrix of dipole moments, and $V_k$ is the $M \times N$ matrix of measurement noise given by

$$V_k = \begin{bmatrix} v_1(1) & v_1(2) & \cdots & v_1(N) \\ v_2(1) & v_2(2) & \cdots & v_2(N) \\ \vdots & \vdots & & \vdots \\ v_M(1) & v_M(2) & \cdots & v_M(N) \end{bmatrix}, \tag{3}$$

where $v_m(t)$ is a zero mean Gaussian noise realization, uncorrelated in time and space, and with variance $\sigma_v^2$ independent of time.

In (2), the lead field matrix $A(\boldsymbol{\theta})$ allows for the calculation of the distributions of the electric potentials given a source configuration. The calculation of those distributions implies solving Maxwell's equations for the physical model of source, realistic head and measurements, with the associated boundary values. There is no known closed-form analytic solution to this *forward problem*, but here we consider the case of the boundary element method (BEM) as it is known to converge to the true solution under certain conditions when the tessellated head model is sufficiently refined [19].

### 2.2. Beamforming and Neural Indices

Based on the measurements in (2), a beamformer $W$ can be proposed in such a way that the dipole's signals could be approximated from the measured data, i.e., $\widetilde{Q}_k = W^T Y_k$. Then, a consistent estimate of $Q$ could be obtained as $\widehat{Q} = \sum_k^K \widetilde{Q}_k / K$ for a sufficiently large value of $K$. Furthermore, the beamformer is usually designed in order to satisfy the following condition:

$$W^T(\boldsymbol{r})A(\boldsymbol{r}) = \begin{cases} I & \text{if} \quad \boldsymbol{r} = \boldsymbol{r}_l \\ \boldsymbol{0} & \text{if} \quad \boldsymbol{r} \neq \boldsymbol{r}_l \end{cases}. \tag{4}$$

In the context of brain signals, the conditions in (4) enforce that $W$ is designed to pass activity coming from a specific location of interest $\boldsymbol{r}_l$, while it attenuates signals coming from other locations $\boldsymbol{r}$, for $\boldsymbol{r} \in \Omega$, where $\Omega$ denotes the volume of the brain.

There are many ways to design a beamformer, but the LCMV approach to achieve optimality is quite effective: to minimize the variance at the filter's output while satisfying the linear response constraint $W^T(\boldsymbol{r}_l)A(\boldsymbol{r}_l) = I$. The solution to such minimization problem may be obtained using Lagrange multipliers (which is the classical method for finding local minima of a function subject to equality constraints) and completing the square, which results in [20]

$$W(\boldsymbol{r}_l)_{\text{LCMV}} = \left[ A^T(\boldsymbol{r}_l)R^{-1}A(\boldsymbol{r}_l) \right]^{-1} A^T(\boldsymbol{r}_l)R^{-1}, \tag{5}$$

where $R = E[YY^T]$ corresponds to the data's covariance matrix. For the case of unknown $R$, a consistent estimate of this covariance matrix (denoted by $\widehat{R}$) can be used.

Furthermore, high variability in the output of the LCMV beamformer as a function of position can be considered indicative of significant neuronal activity [3]. Therefore, to find a source through this approach requires an exhaustive search for the maximum value of the *neural activity index* within $\Omega$. This is written as

$$\hat{r}_l = \max_{r} \frac{\text{tr}\left\{\left[A_l^T(r_l)\widehat{R}^{-1}A(r_l)\right]^{-1}\right\}}{\text{tr}\left\{\left[A_l^T(r_l)\widehat{P}^{-1}A(r_l)\right]^{-1}\right\}} = \max_{r} \iota_{\text{LCMV}}(r), \tag{6}$$

where $\iota_{\text{LCMV}}(r)$, denotes the neural activity index, $\widehat{P}$ is an estimate of the noise covariance matrix, and $\text{tr}\{\cdot\}$ indicates the trace. Note that (6) is equivalent to maximize the source's variance (normalized by the variance of the noise) as a function of $r$.

Since the LCMV filter can only be applied for one source at a time, then the multi-source activity index (MAI) is proposed in [21] as an extension of $\iota_{\text{LCMV}}(r)$ for the case of $L$ neural sources:

$$\iota_{\text{MAI}}(\boldsymbol{\theta}) = \text{tr}\left\{G(\boldsymbol{\theta})H(\boldsymbol{\theta})^{-1}\right\} - 3L, \tag{7}$$

where

$$G(\boldsymbol{\theta}) = A^T(\boldsymbol{\theta})\widehat{P}^{-1}A(\boldsymbol{\theta}), \tag{8}$$

and

$$H(\boldsymbol{\theta}) = A^T(\boldsymbol{\theta})\widehat{R}^{-1}A(\boldsymbol{\theta}). \tag{9}$$

Theoretically, maximizing (7) provides an unbiased estimation of $\boldsymbol{\theta}$ for any signal-to-noise ratio and any level of correlation between the sources. Yet, in practice, these factors strongly affect the spatial resolution of $\iota_{\text{MAI}}$.

### 2.3. Generalized Sidelobe Canceler

A useful implementation of the LCMV beamformer, known as generalized sidelobe canceler (GSC), can be obtained by dividing the data into two subspaces: the *constraint* subspace, which corresponds to the signal components of interest, and an *orthogonal* subspace, in which we find the noise and interference components [20]. A representation of the GSC's structure is shown in Figure 1. The weights of the GSC are given by $W_{\text{GSC}} = W_h - W_0$. There, $W_h$ is the quiescent component of the beamformer and it is given by the solution of (2) for the noiseless condition:

$$W_h = A\left[A^T A\right]^{-1}, \tag{10}$$

hence $\widehat{Q}_h = W_h^T Y_k$. Note that $A = A(\boldsymbol{\theta})$ is used from now on for convenience.

The lower path of the GSC in Figure 1 is comprised by a *blocking matrix*, $C_\perp$, and $W_0$ that corresponds to the projection of $W_{\text{GSC}}$ onto $C_\perp$. Note that the output of the blocking matrix cannot contain any components in the space of $A$, and the design on $C_\perp$ is not unique. One method for constructing it is to find $P_A^\perp = I - A\left[A^T A\right]^{-1}A^T$, then orthonormalize $P_A^\perp$ and choose the first $M - 3L$ columns of the orthonormalized matrix. Such way of designing $C_\perp$ was used in [15] in our first proposal for a specialized neural index based on the GSC.

After applying $C_\perp$ to the measurements, we are left with an estimate of the undesired signals (denoted by $X_0$). Since $W_0$ is the projection of $W_{\text{GSC}}$ onto $C_\perp$, it can be expressed as $W_0 = R_{X_0}^- C_\perp^T \widehat{R} W_h$ where $R_{X_0} = E[X_0 X_0^T]$ and $(\cdot)^-$ indicates the generalized inverse (see [16] for more details).

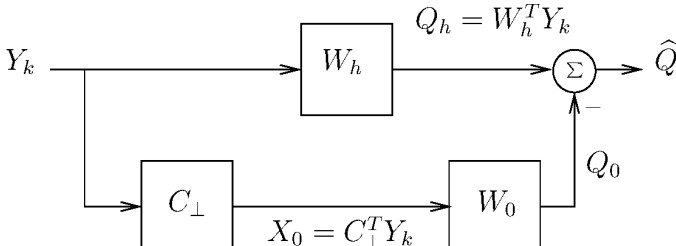

**Figure 1.** Structure of the generalized sidelobe canceler.

### 2.4. Adaptive Blocking Matrix

Our proposed methods rely on the optimal identification and removal of background brain activity and interference and, in the context of the GSC, it is the blocking matrix $C_\perp$ the main responsible of that task. Its quiescent form comes with the disadvantage of not providing an optimal value for the level of rank-reduction. In order to overcome such difficulty, it has been proposed in [17] that such blocking can be optimally designed with the use of an adaptive blocking matrix (ABM). Let us consider the following design for it:

$$C_\perp = \mathrm{I} - \mathbf{b}W_h, \tag{11}$$

where

$$\mathbf{b} = \widehat{R}W_h\left[W_h^T\widehat{R}W_h\right]^-, \tag{12}$$

such that the filters in (12) correspond to the Wiener-Hopf solution that minimize the mean-squares of $X_0$. Then, the ABM in (11) comes from writing $X_0$ as a function of $Y_k$.

### 2.5. Proposed Reduced-Rank Beamforming Scheme

In many practical beamforming applications, the available training data is insufficient to obtain a full-rank estimate of the covariance matrix of interference and noise, additionally the interference is typically of low rank. In [15], we proposed a reduced-rank beamforming approach using the cross-spectral metrics (CSM), since they allow the dimensionality of the filter to be reduced below the dimension of the noise eigenstructure without significant loss of performance in the signal-to-interference-plus-noise ratio (SINR). In addition, the crosspectral reduced-dimension filter can outperform the full-dimension one when the noise covariance is unknown (see [22] and references therein). Under those conditions, we re-evaluate the activity index first proposed in [15]:

$$\iota_{\mathrm{RR}_1}(\boldsymbol{\theta}) = \mathrm{tr}\left\{\left[\widetilde{W}_0^T\widehat{R}^{-1}(\widetilde{W}_0 - W_h) - W_h^T\widehat{R}^{-1}\widetilde{W}_0\right]^{-1}\right\}. \tag{13}$$

This index applies a rank-reduction to $W_0$ (indicated as $\widetilde{W}_0$) according to the $J = M - \mathrm{rank}(C_\perp)$ largest CSM as defined in [23]. Hence, $\widetilde{W}_0$ is given by

$$\widetilde{W}_0 = \sum_{j=1}^{J}\frac{\boldsymbol{u}_j\boldsymbol{u}_j^T R_{X_0Q_h}}{\lambda_j}, \tag{14}$$

where $\lambda_j$ are the eigenvalues and $\boldsymbol{u}_j$ are the orthonormal eigenvectors of $R_{X_0}$ with largest CSM, and $R_{X_0Q_h} = \mathrm{E}[X_0Q_h^T]$. Note that $\eta_{\mathrm{RR}_1}$ differs from the one in [15] because we are now considering $C_\perp$ as an ABM whose rank will determine the rank-reduction of $W_0$.

In (13), all the weights of $W_{\mathrm{GSC}}$ are considered, yet the rank-reduction is only applied to $W_0$. For this reason, we introduce an alternative activity index that only considers the weights of the orthogonal subspace of the noise and interference components:

$$\iota_{\mathrm{RR}_2}(\boldsymbol{\theta}) = \mathrm{tr}\left\{\left[\widetilde{W}_0^T\widehat{R}^{-1}\widetilde{W}_0\right]^{-1}\right\}. \tag{15}$$



Under those conditions, our proposed brain source localization based on the reduced-rank activity indices $\iota_{\mathrm{RR}_1}$ and $\iota_{\mathrm{RR}_2}$ is given by

$$\widehat{\boldsymbol{\theta}} = \min_{\boldsymbol{\theta}} \iota_{\mathrm{RR}_{(\cdot)}}(\boldsymbol{\theta}), \text{ for } \boldsymbol{\theta} \in \Omega. \tag{16}$$

Note that in (13) and (15) a negative sign was omitted for convenience, hence the minimization in (16). In addition, note that their structure is inspired in the one of (6) but with the array response matrix $A$ replaced by a reduced-rank version of the weights $W_{\mathrm{GSC}}$ and $W_0$ in $\iota_{\mathrm{RR}_1}$ and $\iota_{\mathrm{RR}_2}$, respectively. This approach is used to "force" a desired response in a similar way as in array interpolation. There, a virtual or desired array response having all properties necessary for posterior processing is used to provide the array with the necessary characteristics to estimate the direction of correlated sources [24].

### 2.6. EEG Data

In order to prove the applicability of the previously proposed methods, next we present two types of EEG data used in our numerical examples: simulated and real EEG data. In the examples with simulated data, the aim is to evaluate the performance of (7) and compare it to those of (13) and (15) on the inverse problem of estimating the position of $L = 3$ brain sources located on the occipital lobe, specifically in the visual cortex. For the examples with real data, their purpose is to show the applicability of the proposed methods to resolve the topography of cortical activation produced by visual flicker stimulation.

### 2.6.1. Simulated Data

Realistic EEG measurements were simulated using the four-layers geometrical model of the head shown in Figure 2. The measurements were obtained for an array of $M = 90$ sensors. Based on that, we generated $K = 100$ measurements with independent biological noise (i.e., background activity coming from other regions in the brain) and measurement noise realizations, such that (2) became extended to

$$Y_k = A(\boldsymbol{\theta})Q + A(\boldsymbol{\theta}_\Delta)Q_\Delta + V_k, \text{ for } k = 1, 2, \ldots, K, \tag{17}$$

where $\boldsymbol{\theta}_\Delta$ and $Q_\Delta$ are the positions and dipole moments, respectively, of five hundred cortical and one hundred thalamic sources simulating the biological noise. The number of cortical and thalamic sources were chosen to approximate realistic spatially correlated noise and to provide enough power to simulate a desired range of noise conditions (for a discussion on random dipole modeling of spontaneous brain activity, see [25,26]). For both cortical and thalamic sources, their $\|Q_\Delta\|$ were modeled as independent Gaussian random variables with zero mean and variance $\sigma_\Delta^2$. The orientation of each $Q_\Delta$ remained always normal to surface of the triangle in which it was located in the tessellated thalamus. Using (17), and by adjusting the corresponding variances $\sigma_v^2$ and $\sigma_\Delta^2$, we created different scenarios based on the following ratios (in decibels):

- signal-to-measurement-noise ratio, given by

$$\eta_m = 20 \log \frac{\|A(\boldsymbol{\theta})Q\|_{\mathrm{F}}}{\|V_k\|_{\mathrm{F}}}, \tag{18}$$

where $\|\cdot\|_{\mathrm{F}}$ denotes the Frobenius norm;
- signal-to-biological-noise ratio, given by

$$\eta_b = 20 \log \frac{\|A(\boldsymbol{\theta})Q\|_{\mathrm{F}}}{\|A(\boldsymbol{\theta}_\Delta)Q_\Delta\|_{\mathrm{F}}}. \tag{19}$$

Hence, we simulated different sets of measurements with a combination of $\eta_m = 0, 5, 10$ dB, and $\eta_b = -5, 0, 5, 10$ dB (average values over $K = 100$ independent noise realizations).

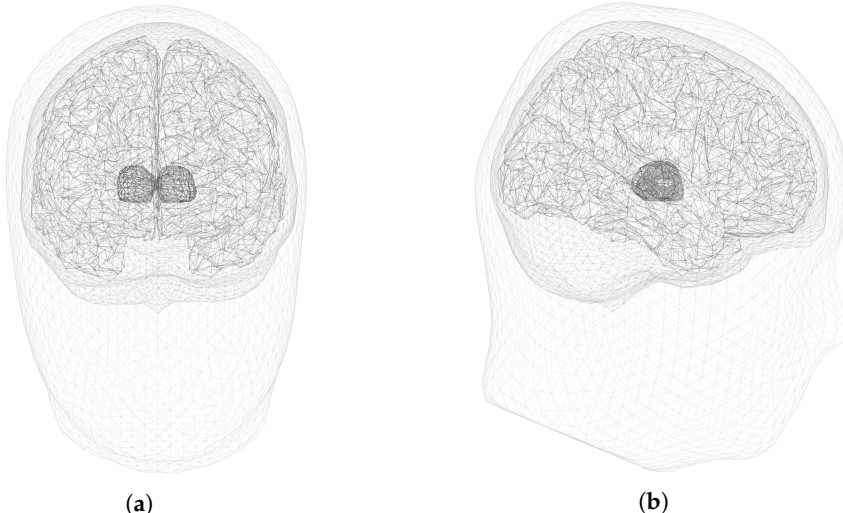

(**a**) (**b**)

**Figure 2.** Geometrical model with four layers (scalp, skull, brain and thalami). (**a**) Frontal. (**b**) Sagittal.

Under those conditions, each trial $k$ was created through (17) to have specific $(\eta_m, \eta_b)$ while keeping $\theta$ and $Q$ unchanged. In all data, we considered the case of $L = 3$ sources located in the occipital lobe and in close proximity to each other in order to generate spatially correlated measurements. For $Q$, their temporal evolution was chosen to model realistic evoked responses. Furthermore, two out of the three dipoles were independent between each other (they are shown in Figure 3). The third dipole was constructed as a linear combination of the other two, i.e.,

$$q_3 = \begin{bmatrix} 0.4 & 0.1 \\ 0.2 & 0.7 \\ 0.5 & 0.9 \end{bmatrix} \begin{bmatrix} q_1 \\ q_2 \end{bmatrix}. \tag{20}$$

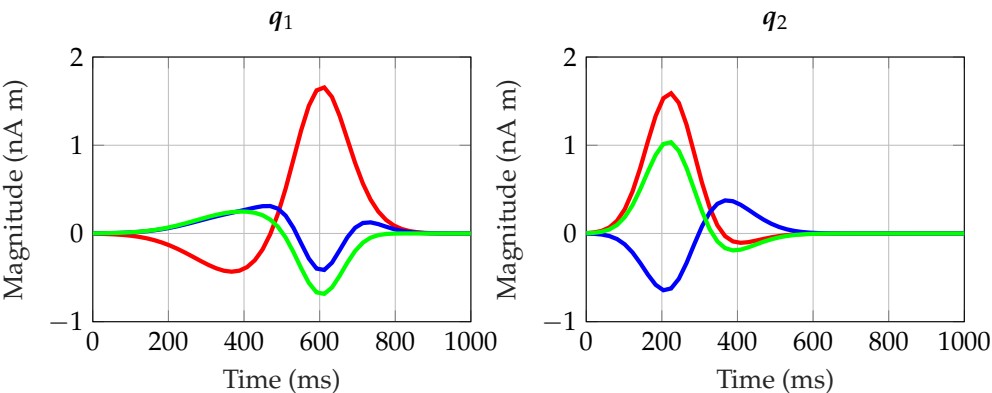

**Figure 3.** Temporal evolution of dipoles $q_1$ and $q_2$. Their Cartesian components $x$, $y$, and $z$ are shown in red, blue, and green, respectively.

This approach was taken in order to further enforce a situation of source correlation in which traditional beamformers are known to fail. Finally, $A(\theta)$ and $A(\theta_\Delta)$ in (17) were calculated by using the comprehensive BEM toolbox that was proposed in [11].

Once we solved the forward problem of data generation, we proceeded with the solution of the inverse problem of estimating $\theta$. For that, all neural indices were computed only within a region-of-interest (ROI) comprising the visual cortex. Therefore, the ROI contained 517 candidate positions out of the 8633 total positions that the tessellated model of the brain cortex had. Next, the pool of candidate positions was reduced by choosing those for which their $\iota_{\text{LCMV}}(r)$ were above the 70-percentile. That process left out all but

150 candidate source positions. Figure 4 shows an example of such ROI including the sources' positions ($\theta$) we used in our simulations.

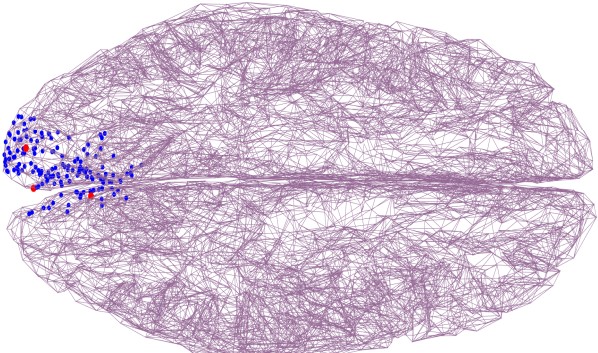

**Figure 4.** Tessellated model of the brain (top view). Red points indicate the $L = 3$ real sources' positions. Other candidate positions that are part of the ROI are shown as blue points.

### 2.6.2. Real EEG Data

In order to prove the applicability of our proposed methods, we performed a dipole source localization using real EEG data corresponding to the activation produced by steady-state visual-evoked potentials (SSVEP). The data we used is freely available in [27].

The stimulus of the experiment was a violet box, presented on the center of the monitor, flickering in five different frequencies (6.66, 7.50, 8.57, 10.00 and 12.00 Hz). The experimental setup is shown in Figure 5. The experiment was divided into five identical sessions, and each one was initiated with 100 s of resting period where the participant could look at the black screen of the monitor without being involved in any activity. A 100 s adaptation period followed, which consisted in the presentation of the five frequencies in a random way. The following 30 s interval was left for the subject to rest and be prepared for the next trial, which consisted in the presentation of one frequency for three times before another 30 s break. Every frequency is presented sequentially for three times and with a resting period of 30 s between each trial. Each session eventually includes 23 trials, with eight of them being part of the adaptation [28].

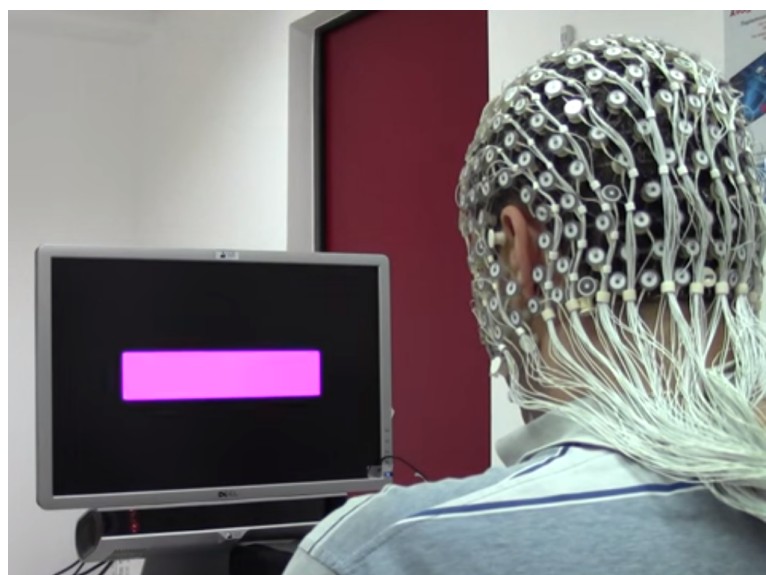

**Figure 5.** Experimental setup for the acquisition of EEG SSVEP data. Image taken from https://youtu.be/8lGBVvCX5d8 (accessed on 2 February 2023).

## 3. Results

This section shows the results of our numerical examples using the data described in Sections 2.6.1 and 2.6.2, respectively.

### 3.1. Evaluation of Performance under Different $\eta_b$ and $\eta_m$ Conditions

Based on the realistically simulated EEG data, the estimation process was performed for all the possible combinations of three candidate positions from our pool (up to 551,300 in our case). Hence, the indices in (7), (13) and (15) were computed for $\boldsymbol{\theta}_{\text{cand}} = [\boldsymbol{r}_{c1}, \boldsymbol{r}_{c2}, \boldsymbol{r}_{c3}]^T$, with $\{\boldsymbol{r}_{c1}, \boldsymbol{r}_{c2}, \boldsymbol{r}_{c3}\} \in \text{ROI}$. The estimated parameter $\hat{\boldsymbol{\theta}}_k$ for MAI corresponded to the value of $\boldsymbol{\theta}_{\text{cand}}$ for which (7) was maximum, while the estimated parameter for our proposed indices where those according to (16).

Next, we used the following metrics to evaluate the performance of all the estimations:

- the individual bias of the estimates, given by $b_{l,k} = ||\boldsymbol{r}_l - \hat{\boldsymbol{r}}_{l,k}||_2$;
- their sum-of-squares: $\text{SS}_k = \sum_{l=1}^{3} b_{l,k}^2$;
- the maximum bias: $b_{\text{MAX},k} = \max_l b_{l,k}$.

The results are finally presented for different combinations of $\eta_b$ and $\eta_m$ as an average over the $K = 100$ trials. Hence, we compared the average sum-of-squares (denoted as $\overline{\text{SS}}$), and the standard deviation of the maximum bias (denoted as $\sigma_{b_{\text{MAX}}}$). The first provides us with an overall view of estimation errors as it adds up the squared-bias in the estimation of the three sources, while the second give us the worst case scenario, i.e., the largest variability in the estimation of each of the sources. These results are shown in Figures 6 and 7, respectively. Note that both $\overline{\text{SS}}$ and $\sigma_{b_{\text{MAX}}}$ are dimensionless as we normalized them against the mean node distance of tessellated model of the brain cortex, as the minimum spatial resolution that we could achieve in our simulations was constrained by the size of the triangles in our mesh. Additionally, note that in those figures we have included (only for comparison purposes) the result of a source localization based only on minimizing the power of $Q_0$ as a function of $\boldsymbol{\theta}$, which is denoted as $P(W_0)_{\text{min}}$.

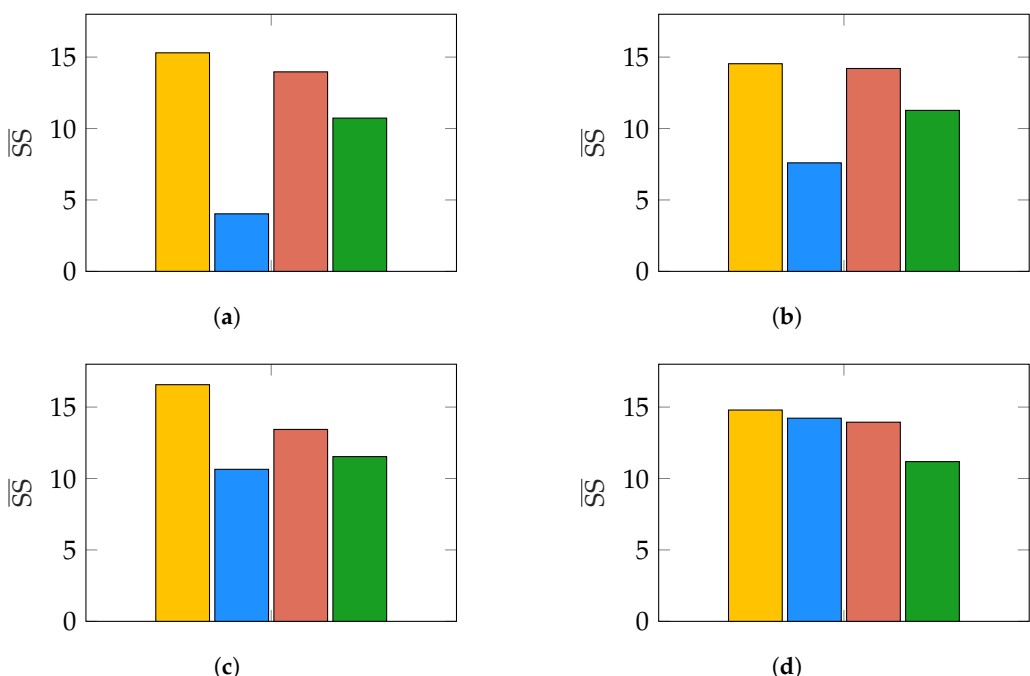

**Figure 6.** Results of evaluating $\overline{\text{SS}}$ for different noise conditions. Bars correspond to $P(W_0)_{\text{min}}$ (yellow), $\iota_{\text{MAI}}$ (blue), $\iota_{\text{RR}_1}$ (red), and $\iota_{\text{RR}_2}$ (green). (**a**) Noise conditions: $\eta_b = 10$ dB, $\eta_m = 10$ dB; (**b**) noise conditions: $\eta_b = 5$ dB, $\eta_m = 5$ dB; (**c**) noise conditions: $\eta_b = 0$ dB, $\eta_m = 0$ dB; (**d**) noise conditions: $\eta_b = -5$ dB, $\eta_m = 0$ dB.

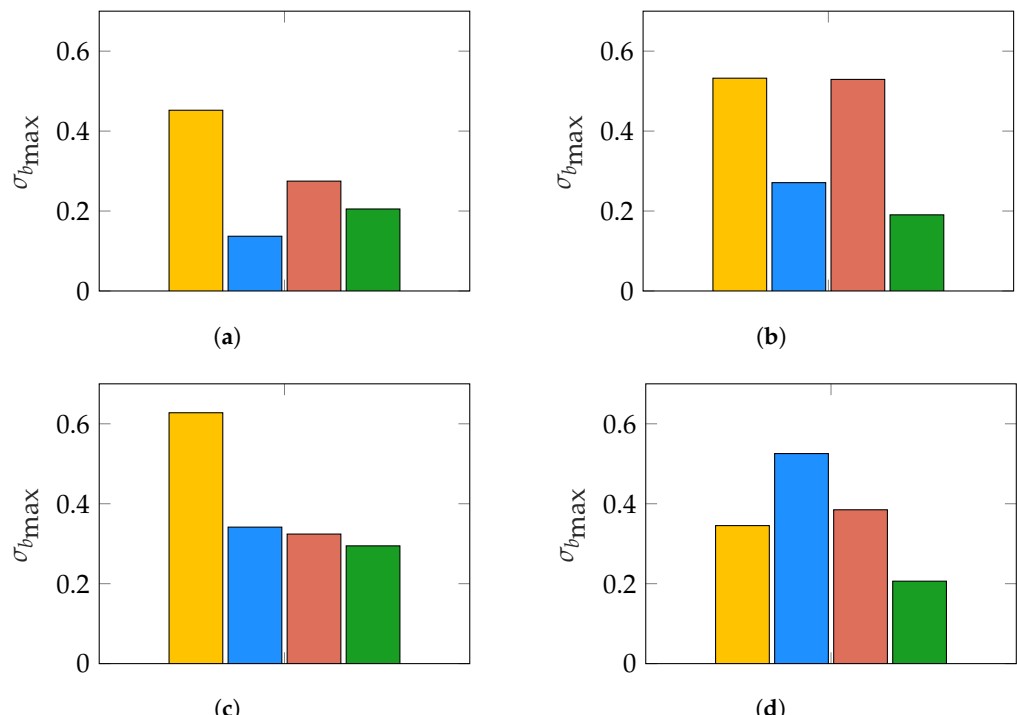

**Figure 7.** Results of evaluating $\sigma_{b_{max}}$ for the same noise conditions as in Figure 6. Bars correspond to $P(W_0)_{min}$ (yellow), $\iota_{MAI}$ (blue), $\iota_{RR_1}$ (red), and $\iota_{RR_2}$ (green). (**a**) Noise conditions: $\eta_b = 10$ dB, $\eta_m = 10$ dB; (**b**) noise conditions: $\eta_b = 5$ dB, $\eta_m = 5$ dB; (**c**) noise conditions: $\eta_b = 0$ dB, $\eta_m = 0$ dB; (**d**) noise conditions: $\eta_b = -5$ dB, $\eta_m = 0$ dB.

The results in Figure 6 show that $\overline{SS} \approx 10$ in the case of $\iota_{RR_2}$ for all $(\eta_m, \eta_b)$ pairs we evaluated. Furthermore, we noticed in our experiments that the most common case was that of two dipoles correctly estimated while the third was off the true value of $\mathbf{r}_l$, hence accounting for most of the bias. Under that consideration, the average error in the estimation is as large as 3.16 times the mean-node distance of our mesh modeling the brain for $\iota_{RR_2}$. As expected, $\iota_{MAI}$ keeps a lower error only in high $\eta_b$ (Figure 6a,b), but the spatially correlated biological noise rapidly degrades its performance as $\eta_b$ gets lower. The standard deviation of the worst-case scenario is shown in Figure 7. There, $\sigma_{b_{max}} \approx 0.2$ times the mean-node distance for $\iota_{RR_2}$. In the case of $\iota_{MAI}$, this value is only lower than the one of our proposed index for the best conditions $(\eta_m, \eta_b) = (10$ dB, 10 dB$)$. For all other conditions, $\sigma_{b_{max}}$ increases up to 0.525 times the mean-node distance. Clearly, $\iota_{RR_2}$ provides a biased solution with low variability for all noise conditions, regardless of time and spatial correlation between the sources.

### 3.2. Applicability of $\iota_{RR_1}$ and $\iota_{RR_2}$ in Dipole Source Localization Using Real EEG Data

We implemented the dipole source localization process using (13) and (15) in the data of two different subjects. For the purpose of this test, we consider the search of $L = 3$ sources as in the case of Section 2.6.1. Since the trials were presented three times for the five selected frequencies, we were left with a 15 s period of stimulation for each frequency which we divided into shorter processing windows of 1 s. Since the data provided no anatomical information of the subjects, we approximated them by using the same head model as in Section 2.6.1. This process required the sensor locations to be adjusted (morphed) to our head model. Finally we evaluated our neural indices $\iota_{RR_1}$ and $\iota_{RR_2}$ in the ROI previously defined.

The results of our source localization are shown in Figures 8 and 9 in the way of activation maps over the cortex. There, the lower values of $\iota_{RR_1}$ and $\iota_{RR_2}$ are shown in darker colors. Even though a three-dipole search was considered, the maps represent $\iota_{RR_1}(\boldsymbol{\theta}_{cand})$ and

$\iota_{\mathrm{RR}_2}(\boldsymbol{\theta}_{\mathrm{cand}})$, i.e., the maps show the average activity index over 15 processing windows for each candidate position in the ROI. Since a single $\boldsymbol{r}_{cl}$ can be part of multiple combinations of $\boldsymbol{\theta}_{\mathrm{cand}}$, those index values were also accounted and averaged. Under those conditions, our results show very distinctive activation over primary visual cortex V1 (right for Subject 5 and left for Subject 10) in the case of $\iota_{\mathrm{RR}_1}$, while activation in secondary visual cortex V2 are more noticeable in the case of $\iota_{\mathrm{RR}_2}$ and in the same hemispheres as in the case of the activation of V1 for both subjects. Given the results of Section 3.1 in which $\iota_{\mathrm{RR}_2}$ showed less bias and variability than $\iota_{\mathrm{RR}_1}$, we believe that activation of V2 could be unnoticeable for $\iota_{\mathrm{RR}_1}$ due to estimation errors.

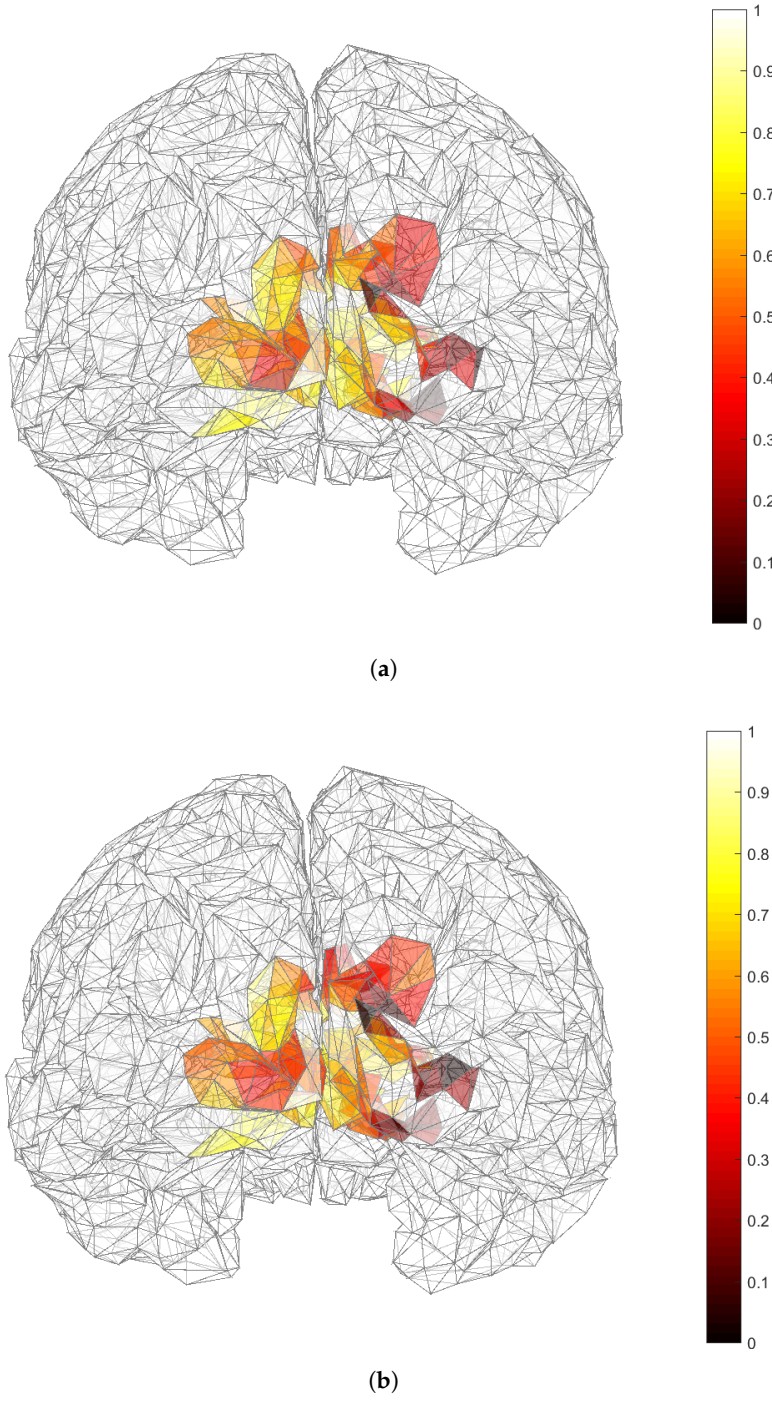

**Figure 8.** Results of the source localization process through our reduced-rank beamforming approach in EEG SSVEP data of Subject 5 in [27] and for a specific frequency. Minimum values of $\iota_{\mathrm{RR}_{(\cdot)}}$ (shown in darker colors) correspond to zones of main brain activation. (**a**) $\iota_{\mathrm{RR}_1}$ (**b**) $\iota_{\mathrm{RR}_2}$.

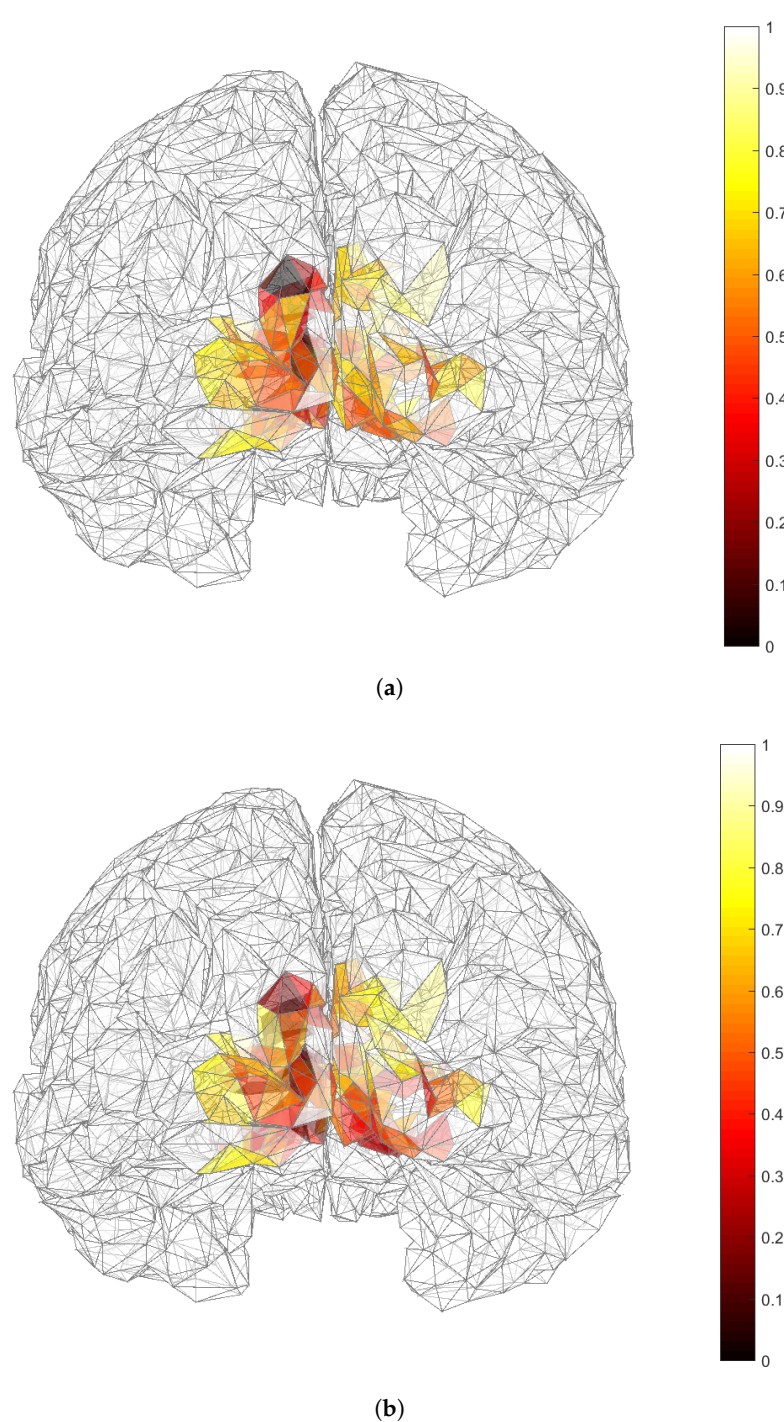

**Figure 9.** Results of the source localization process through our reduced-rank beamforming approach in EEG SSVEP data of Subject 10 in [27] and for a specific frequency. Minimum values of $\iota_{RR_{(\cdot)}}$ (shown in darker colors) correspond to zones of main brain activation. (**a**) $\iota_{RR_1}$ (**b**) $\iota_{RR_2}$.

## 4. Discussion

It is known that LCMV-based solutions may perform sufficiently well when solving the neuroelectric inverse problem of estimating the location of brain sources only if certain conditions are satisfied: zero or low correlations between the sources, high enough signal-to-noise ratios (SNRs), and sufficiently large spatial separation of sources [8]. However, the LCMV beamformer tends to suppress source-power estimates from sources that have highly correlated time-courses, given that it assumes that source time-courses from different generators are uncorrelated, which cannot always be attained [3,29].

In beamforming applications, the available training data is insufficient to obtain a full-rank estimate of the interference-and-noise covariance matrix mainly because the interference is typically of low rank, therefore minimum-variance filtering methods in low-rank interference have attracted considerable attention [16]. In [8], a reduced-rank version of $\iota_{\text{MAI}}$ was presented, in which spatial resolution compared with its full-rank version was improved. However, the selection of the optimal level of rank-reduction is still an issue that requires to be solved. Therefore, in this paper we centered our comparisons around the original neural index proposed in [21], and we avoided directly entering in the problem of optimal rank selection for our own indices. Instead, we trusted the optimality conditions provided by the solution of the Wiener-Hopf for the calculation of the ABM. Yet, we recognize the issue requires further investigation, but it is out of the scope of this paper.

With respect to our results using simulated data, they indicate that our proposed neural indices are not greatly affected by an increase in $\eta_b$ and $\eta_m$, while the performance of MAI degrades with the increase of noise. The consistency in the response of our proposed beamformers is due to their specific design, in which optimality is achieved in the estimation (and later cancellation) of the noise and interference components. Such affinity to the orthogonal space in the GSC of those undesired components is achieved both by the ABS and the rank-reduction based on the CSM. An strategy in which no rank-reduction is applied turns to be inefficient, as the results with $P(W_0)_{\text{min}}$ show.

For the case of real data, we used brain sources related to SSVEP. Our results under those conditions show that our methods are capable of localizing brain sources in the visual cortices V1 and V2, which are expected for the SSVEP. In [30], by using functional magnetic resonance imaging (fMRI) and dipole modeling, they were able to analyze the neural generators of the SSVEP. They found that the primarily origin for these stimulus came from two concurrently active dipolar sources that were situated in medial and lateral occipital cortex, providing strong evidence that phase-locked neural activity in area V1 is a major contributor to the pattern-reversal SSVEP and in some of the single subjects fMRI also showed activity in area V2, but due to their close proximity these were included in the V1 sources. Our results are in agreement with those findings, as in both subjects we were able to map brain activation in V1 and V2. Nevertheless, our results have the uncertainty produced by fitting a head model that did not corresponded to the subjects. Still, our results make sense in term of the physiological event, even when they were estimated as average responses over only 15 processing windows.

Although the mechanism of SSVEP is not quite clear, their generation via the visual pathway starts when the light projects on the rod cells or cone cells in the retina generating electronic signals then this signals are transferred to the ganglion cells to be pre-processed and transferred to the V1. There, the response of the neurons represents the integrated signals in an area or even a whole visual field. Finally, the signals integrated in the V1 are transferred to other areas in the brain for further processing. All the previous characterization is important because the neural networks associated to SSVEP can overlap with parts of the visual recognition network. Hence, some researchers believe that combining SSVEP extraction method with the superposition method is a valid way for elucidating the process of a recognition task over a long duration of time [31]. It is here where our proposed methods have the potential of helping, as our results show, due to the capacity of beamforming techniques to focalize the search in specific areas of interest. This is perhaps not fully demonstrated in our results given that we depended on a public database and we had limited access to information related to the frequencies of stimulation. For that reason, future work in this area will include a more exhaustive investigation with SSVEP data we plan to acquire ourselves.

One important use for the SSVEP is to build an SSVEP-based brain-computer interface (BCI) which can measure brain activity via invasive or non-invasive means, relying on different mental strategies that each produce brain activity patterns that can be detected by a pattern recognition system [32]. SSVEP-based BCIs utilize different frequency flickers to represent different tasks, then a subject can complete a task by simply staring at the

flicker that corresponds to that work. Therefore, the intention of the subject can be decoded by recognizing the SSVEP frequency in the evoked EEG [31]. In that sense, our results show that the proposed methods can be helpful with this task, as differences in the brain activation can be related to different frequencies, as it can be seen in Figures 8 and 9.

## 5. Conclusions

We proposed two neural indices for the problem of dipole source localization. These indices were designed specifically to suppress interference that are commonly present in EEG data and are often considered as "background" activation. Our approach comes from a combination of two different processing elements that are implemented under the structure of the GSC in order to improve the estimation and later cancellation of noise and interference, regardless of the SINR. Those elements are (i) an ABM that achieves an optimal estimation of unwanted signals and defines the rank-reduction that (ii) a CSM-based approach will apply to the filter to improve the affinity to the noise subspace. We showed the applicability of our methods for dipole source localization using real data in which LCMV-based methods could not be used given the ill-conditioned nature of the problem. We also showed that our beamformers are capable of estimating real sources from SSVEP.

There are many factors that can affect the accuracy of the EEG source localization methods, such as the effects of the head cavities, variations in tissue conductivities, or errors in the position of the electrodes. Hence, in addition to improve the localization methods with techniques such as those we proposed here, other strategies need to be placed in practice to reduce the adverse effects previously mentioned. Even beamforming techniques such as the ones proposed here suffer a fundamental limitation: their performance directly depends on the number of sensing elements (aperture), independently of the number of time samples or the signal-to-noise ratio. Again, it is only through overlapping approaches that non-invasive localization methods of the sources in the brain can be used to diagnose pathological, physiological, mental, and functional abnormalities such as the study of localized epilepsy, evoked-related-potentials or attention deficit/hyperactivity disorder [1].

**Author Contributions:** Conceptualization, D.G.; Data curation, E.J.-C.; Formal analysis, E.J.-C.; Investigation, E.J.-C.; Methodology, E.J.-C. and D.G.; Project administration, D.G.; Resources, E.J.-C.; Software, E.J.-C. and D.G.; Supervision, D.G.; Validation, E.J.-C. and D.G.; Visualization, D.G.; Writing—original draft, E.J.-C.; Writing—review and editing, D.G. All authors have read and agreed to the published version of the manuscript.

**Funding:** The work of Eduardo Jiménez-Cruz was sponsored by the Mexican Council of Science and Technology (CONACyT) through the graduate school scholarship no. 727361.

**Institutional Review Board Statement:** The real data used in this paper was acquired in accordance with the Declaration of Helsinki, and approved by the Ethics Committees of the Third Neurological Clinic of the Aristotle University of Thessaloniki, Muscular Dystrophy Association Hellas, and Spinal Cord Injury Rehabilitation Center of Sheba Medical Center, all in Greece.

**Informed Consent Statement:** Informed consent was obtained from all subjects involved in the study.

**Data Availability Statement:** The data sets used in this study are available under request by contacting the corresponding author.

**Conflicts of Interest:** The authors declare no conflict of interest.

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
