# Peer review of "Interference Suppression in EEG Dipole Source Localization through Reduced-Rank Beamforming"

_applsci, doi:10.3390/app13053241_

Round 1

Reviewer 1 Report

This paper studies the interference suppression in electroencephalography (EEG) dipole source. The topic is timely and suitable for publication in Applied Science. Some weak points are given below:

It is not clear how the simulated data is generated.

The reviewer suggests explicitly presenting both the signal-to-measurement-noise ratio and the signal-to-biological-noise ratio in the mathematical frameworks.

The comparisons with other methods are missing too. The reviewer believes that if the comparisons with state-of-the-art are given, the contributions of the present paper will improve significantly.

The authors should provide a table to summarize all variables, and notations used in the paper.

There still have typos in the manuscript, please carefully proofread the paper before submitting it.

Author Response

We would like to thank the reviewer for all the comments. The revised version of the manuscript has addressed most of the concerns of the reviewer. Please see attachment for a document showing the changes in the manuscript. Next, we have a point-by-point response to the reviewer's comments: 

>>> It is not clear how the simulated data is generated.

Reply: More details on how simulated data is generated has been added to Section 2.6.1

>>> The reviewer suggests explicitly presenting both the signal-to-measurement-noise ratio and the signal-to-biological-noise ratio in the mathematical frameworks.

Reply: Formal mathematical definitions for both ratios have been added in Section 2.6.1 as well

>>> The comparisons with other methods are missing too. The reviewer believes that if the comparisons with state-of-the-art are given, the contributions of the present paper will improve significantly.

Reply: Most of the methods for solving the inverse problem of dipole source localization are known to underperform in conditions such as those we are addressing in this paper. Only the multi-source activity index (MAI) provides, in theory, an unbiased estimate, yet in practice its performance is degraded by temporal and spatial correlations. For this reason, we decided to compare against the best method in theory. Other options that has been recently proposed is a reduced-rank version of MAI, which corresponds to [8] in the list of references (revised manuscript). However, we decided not to use it in this paper as a comparison against that index would be unfair, given that the issue of optimal rank selection has not yet been elucidated. In our case, we decided to use a Wiener filter for that purpose, still a full optimality analysis is required, but out of the scope of our paper. Finally, we currently do not count with access to high-end computer resources and, with our current capabilities, a series of new experiments would have taken months to finish. Therefore, we believe our current experiments provide with fair comparisons and help to prove our point about the specific case of interference suppression. 

>>> The authors should provide a table to summarize all variables, and notations used in the paper.

Reply: This table has been added at the beginning of Section 2.

>>> There still have typos in the manuscript, please carefully proofread the paper before submitting it.

Reply: The manuscript underwent a style and spell check as recommended. 

Reviewer 2 Report

The article touches on an interesting topic. This is because it seems that EEG research already has enough well established that little new instruction can be added to it. The proposed solution has great potential for interdisciplinary use.

Author Response

We would like to thank the reviewer for the nice comments. Please see attachment for a document showing the changes in the manuscript.

Reviewer 3 Report

The authors propose indices intended for the solution of the inverse problem related to EEG source localization, which is a research field particularly dear to the EEG community.

The manuscript is overall well written and the methodology explained throughly by also referencing to pertinent previous publications of the authors. More space should be given to the discussion of the obtained results.

Therefore, I suggest a major revision, thinking that the manuscript and the readership will benefit from a complete insight on the achieved results.

Major comments:

- The title presents a typo: supression > suppression.
- Line 194: How do the authors estabilish the number of cortical and thalamic sources?
- Figure 5 and 6 should be explained clearly and a precise interpretation of the results given.
- Line 253: An in depth interpretation of Figure 7 and 8 should be provided.

Minor comments:
- There are some typos in the text, e.g. line 6, 8, 100, 280, 312, and 337.
- The authors could include more up to date citations.

Author Response

We would like to thank the reviewer for all the comments. The revised version of the manuscript has addressed most of the concerns of the reviewer. Please see attachment for a document showing the changes in the manuscript. Next, we have a point-by-point response to the reviewer's comments:

>>> The title presents a typo: supression > suppression.

Reply: This one, as well as other typos, have been corrected.

>>> How do the authors establish the number of cortical and thalamic sources?

Reply: Full details are now provided in Section 2.6.1 (Line 197 of revised manuscript)

>>> Figure 5 and 6 should be explained clearly and a precise interpretation of the results given.

Reply: We have added more details on the results regarding Figs. 5 and 6 (starting in Line 270 of revised manuscript).

>>> An in depth interpretation of Figure 7 and 8 should be provided.

Reply: We have added more details on those results as well (starting at Line 293 of revised manuscript)

>>> There are some typos in the text, e.g. line 6, 8, 100, 280, 312, and 337.

Reply: The manuscript underwent a style and spell check as recommended.

>>> The authors could include more up to date citations.

Reply: We were able to update one citation, and added two more. It is difficult not to cite old papers in this topic as it is only recently that it is regaining interest.

Round 2

Reviewer 3 Report

I thank the Authors for having addressed my concerns and for presenting such an interesting methodology for EEG source localization to the EEG research community.

I suggest manuscript acceptance in the present form.